# Strategies for the Prevention of the Intra-Hospital Transmission of COVID-19: A Retrospective Cohort Study

**DOI:** 10.3390/healthcare8030195

**Published:** 2020-07-03

**Authors:** Min Cheol Chang, Jian Hur, Donghwi Park

**Affiliations:** 1Department of Rehabilitation Medicine, College of Medicine, Yeungnam University, Daegu 38541, Korea; wheel633@ynu.ac.kr; 2Department of Infectious Disease Internal Medicine, College of Medicine, Yeungnam University, Daegu 38541, Korea; 3Department of Physical Medicine and Rehabilitation, Ulsan University Hospital, College of Medicine, University of Ulsan, Ulsan 44610, Korea

**Keywords:** COVID-19, hospital infection, secondary infection, mask, isolation

## Abstract

Coronavirus disease (COVID-19) has spread rapidly worldwide. We aimed to review the strategies used by our university hospital in Daegu (South Korea) to prevent the transmission of COVID-19 within our institution. We also investigated the actual situation at our hospital against the recommended guidelines. We conducted a survey among patients and staff in our hospital. Additionally, patients’ electronic medical records were reviewed along with closed-circuit television (CCTV) recordings. Various strategies and guidelines developed by our hospital have been implemented. A total of 303 hospital staff and patients had exposure to 29 confirmed COVID-19 patients. Of them, three tested positive for COVID-19 without further transmission. The intra-hospital infection of the disease occurred when the recommended strategies and guidelines such as wearing a mask and isolating for 2 weeks were not followed. In conclusion, the implementation of robust guidelines for preventing the intra-hospital transmission of COVID-19 is essential.

## 1. Introduction

In early December 2019, an increasing number of pneumonia cases due to a novel coronavirus (SARS CoV-2) were reported in Wuhan, China. Since then, the coronavirus disease (COVID-19) has spread not only within China but also to more than 200 countries within 3 months [1,2]. In South Korea, the first imported case of COVID-19 was reported at the end of January 2020 [3]. In mid-February 2020, a big outbreak of COVID-19 occurred due to mass infection at the Shincheon-ji Church in Daegu [3]. As of 23 April 2020, 63.8% of Korean patients with COVID-19 were infected at Daegu [4], and the prevalence rate per 100,000 people of Daegu was 280.98 [5]. The community transmission of COVID-19 in Daegu has increased the risk of intra-hospital transmission. Therefore, when patients who are confirmed or highly suspected of having COVID-19 visit or are admitted to hospital, it is vital that they are isolated as soon as possible to minimize the likelihood of nosocomial spread. However, taking into account asymptomatic patients with COVID-19 and the difficulties distinguishing between patients who have COVID-19-related respiratory symptoms and non-COVID-19-related respiratory symptoms caused by other viruses, preventing the transmission of COVID-19 can be challenging [6]. To minimize intra-hospital transmission, our hospital—one of the four university hospitals in Daegu—established many prevention strategies and guidelines.

In this study, we reviewed the strategies and the guidelines used by our hospital to prevent the transmission of COVID-19 among the patients and hospital staff members. Furthermore, we investigated the actual situation at our hospital.

## 2. Methods

This study was approved by the Institutional Review Board of Yeungnam university hospital (number. 2020-03-014). We reviewed the strategies used at Yeungnam University Medical Center, one of the four university hospitals in Daegu, for preventing the spread of COVID-19 within the hospital during the COVID-19 outbreak in Daegu. Additionally, we reviewed the actual situation at Yeungnam University Medical Center; initially, we conducted a questionnaire survey with patients and staff who came into contact with patients with COVID-19. We double checked the results of the survey using patients’ electronic medical records and closed-circuit television (CCTV) recordings. The following information was investigated: Did they stay away from the confirmed case by at least 3 m? Were they exposed for more than 15 min? Were they wearing a mask at the time of contact? What type of mask was worn? Were they wearing a face shield, AP gown, gloves, and Level D gown? Were they isolated? Did they undergo a COVID-19 test a few days after the exposure and what were the results? Did they have a COVID-19 test on the 13th day after the exposure and what were the results? 

## 3. Results

### 3.1. Strategies for Preventing Intra-Hospital Spread of COVID-19

A total of 29 patients with COVID-19 visited or were admitted to our hospital between 17 February and 11 April 2020, without knowing that they were infected with SARS-CoV-2 (Figure 1). To prevent the spread of COVID-19, Yeungnam University Medical Center developed many guidelines for patients, guardians, and medical staff (Table 1 and Table 2).

The places where patients visited or stayed include the outpatient department (11 patients), ward (9 patients), emergency room (ER) (7 patients), and administration department (facility management team) (2 patients). According to the standard that was announced by the Korea Centers for Disease Control and Prevention (KCDC) on 22 February 2020 (Table 3), staff members would have a low infection risk if both the confirmed patient and the exposed staff member wear a mask. Accordingly, such staff members were informed to “self-monitor” for 2 weeks to check whether they developed a fever and/or respiratory symptoms. If none were observed, then the exposed staff member could stop monitoring their symptoms. If both the confirmed patient and the exposed staff member were not wearing a mask, then the staff member was “isolated” for 2 weeks and required to undergo a COVID-19 test. If they tested negative for COVID-19, then they were allowed to return to work. In addition, patients who came into contact with a confirmed patient were isolated in a single occupancy room for 2 weeks during the hospitalization period. If they presented with fever or respiratory symptoms during the “isolated” period, a COVID-19 test was performed. The wards for patients who contacted a confirmed patient, patients who were confirmed COVID-19 patients, and general patients not related to COVID-19 were separated.

### 3.2. Actual Situation in Our Hospital during the COVID-19 Outbreak

A total of 257 hospital staff members and 46 patients (total 303 individuals) were exposed to 29 confirmed COVID-19 patients (Figure 1), including 186 people in the ward, 70 people in the ER, 40 people in the outpatient department, and 7 people in other places. Among those exposed, 113 were nurses, 64 were doctors, 18 were auxiliary nurses, 17 were radiological technologists, 9 were medical laboratory technologists, 7 were nursing supports, 7 were supporting staff members, 4 were office assistants, 3 were sanitary workers, 3 were emergency medical technicians, 3 were food-cart deliverers, 2 were security guards, 1 was an office worker, 1 was a facility management staff member, 1 was a trainee, 1 was a speech therapist, 1 was a nutritionist, 1 was a crossing guard, 1 was an administration staff member, and 46 were patients (Appendix A). 

Among the 303 hospital staff members and patients who were exposed to the confirmed COVID-19 cases, 57 (18.8%) came into contact with a confirmed patient within a 3 m distance, 94 (31.0%) were in contact for more than 15 min, and 291 (96.0%) wore a mask at the time of contact (4 with N95 masks, 123 with KF94 masks, and 165 with dental masks). Face shields, AP gowns, gloves, and level D gowns were worn by 10, 36, 30, and 3 people, respectively. Of the 303 people, 115 (38.0%) were isolated, and 188 (62.0%) were subjected to self-monitoring. A few days after their contact with the confirmed patient, 161 (53.1%) were tested for COVID-19, of which 3 were confirmed as positive. Subsequently, 71 people (23.4%) underwent testing on the 13th day after exposure, of which no further people were confirmed as positive for SARS-CoV-2. 

### 3.3. Transmission Path of COVID-19

Of the 303 hospital staff members and patients who came into contact with a confirmed patient, three people had a positive result for the presence of COVID-19 (Figure 1, Appendix A). A confirmed patient (Case 5) came into close contact with another confirmed patient (Case 2) in the ER. At that time, only one confirmed patient (case 2) had COVID-19. Both people were not wearing a mask when they came into contact with each other. The confirmed patient in Case 5 was initially isolated in a single occupancy room. After 7 days, the patient strongly requested to move to a multiple-occupancy room. As the patient had initially a negative result for COVID-19 and the average incubation period of COVID-19 is 4.3 days, the patient was allowed to transfer to a multiple-occupancy room. On the 14th day, the patient had fever and was confirmed to have COVID-19 after testing. According to the Management Rule for confirmed cases among inpatients of Yeungnam University Medical Center (Table 3 and Table 4), the patients who were in the same ward were isolated in single-occupancy rooms. Among those isolated, two patients (Cases 5–11 and Cases 5–14, Appendix A) were confirmed to have a positive result. However, both patients wore a mask, except during mealtime, while they were in the same ward as that of the confirmed patient (Case 5).

## 4. Discussion

In this study, a total of 303 hospital staff members and patients were exposed to 29 confirmed COVID-19 patients. Of the 303 exposed individuals, three tested positive for the presence of COVID-19 without further transmission. Various strategies were developed and implemented by our hospital to minimize the in-hospital transmission of COVID-19. Considering the high transmission rate of COVID-19 [7], intra-hospital transmission seems to have been well controlled, as only three patients were found to have been infected through intra-hospital transmission. 

One (Case 5; Appendix A) of the three confirmed patients is believed to have been infected through infected droplets after not wearing a mask while in contact with a confirmed COVID-19 patient (Case 2). The other two patients were infected because our hospital failed to adhere strictly to the 2-week isolation guidelines due to the patient’s request to move to a multiple-occupancy room. Therefore, the hospital’s guidelines that aimed to prevent the spread of COVID-19 were not followed. Thus, these three cases highlight the need for the robust implementation of the guidelines to prevent in-hospital transmission. 

SARS CoV-2 has been reported to spread mainly through infected droplets [1,8]. Infection in droplets is considered to occur when large droplets (>5 µm in diameter) that carry the infectious agent become an infection source [1,8,9]. The droplets tend to be heavy and are present around the patient (within 2 m in indoor air); hence, transmission mainly occurs between close contacts. The size of the coronavirus is approximately 0.12–0.15 µm [9,10]. Therefore, it is not easily filtered except with an N95 mask, which can filtrate particles 0.1–0.3 µm in size [9,11]. However, wearing another type of mask such as a dental or surgical mask has been reported to confer protection against viral transmissions, because the mask can still filter droplets containing virus particles [12,13,14]. Surgical masks are also designed to filter 99.9% of aerosols that are 3 µm in size [12,13,14]. Therefore, N95 masks, KF94, and dental masks can all filter large droplets as well as a significant number of aerosols. Considering that a significant amount of viral penetration is required for viral infection, it is believed that if both clinicians and patients are wearing a mask then the risk of COVID-19 transmission is reduced. 

In a previous study, the median incubation period of COVID-19 was estimated to be 5.1 days (95% CI, 4.5 to 5.8 days) [15] of those who develop symptoms, 97.5% will do so within 11.5 days (95% CI, 8.2 to 15.6 days) [15]. Therefore, if an individual has a history of contact with a confirmed COVID-19 patient, isolation for at least 2 weeks and careful observation may help to minimize the intra-hospital transmission of COVID-19. 

## 5. Conclusions

In conclusion, we showed the efficaciousness of the strategies implemented in preventing the intra-hospital transmission of COVID-19 by reviewing the strategies themselves and the actual situation at our hospital. We hope that our strategies and experience will help to minimize in-hospital infections and assist other hospitals in coping with the COVID-19 pandemic.

## Figures and Tables

**Figure 1 healthcare-08-00195-f001:**
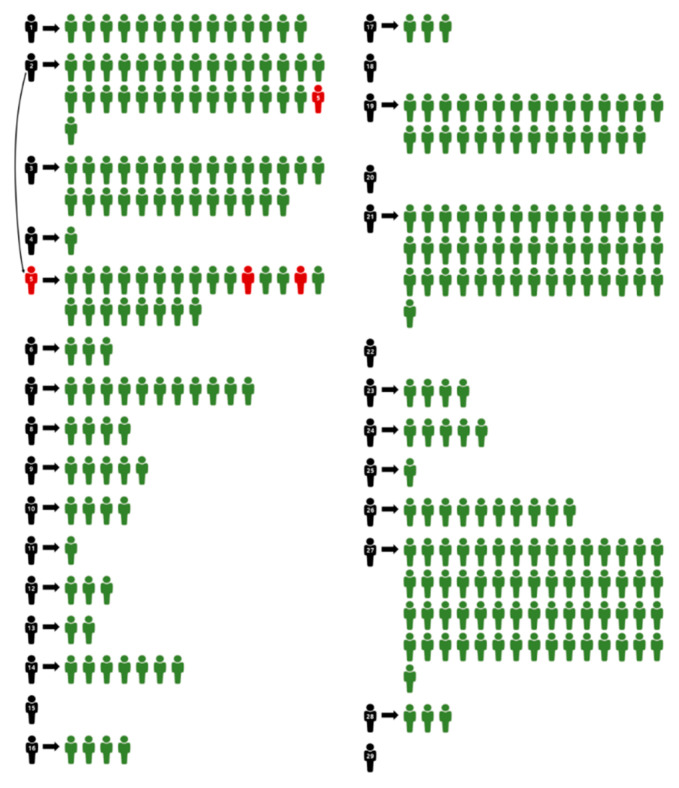
Distribution of the confirmed COVID-19 population and individuals who came into close contact with a COVID-19 patient. Black: confirmed COVID-19 patients; red: infected after coming into contact with a confirmed COVID-19 patient; green: not infected after coming into contact with a confirmed COVID-19 patient. *Case 15, 18, 20, 22, and 29 were isolated immediately after they arrived at the hospital because they had symptoms related to COVID-19.

**Table 1 healthcare-08-00195-t001:** Rules for patients and guardians to prevent nosocomial COVID-19 infection.

Patients and guardians must wear a mask before moving ward or moving around their room.
2. Guardians should not stay in the hospital unless they state a significant reason for the stay.
3. The number of guardians allowed is limited to 1 per patient.
4. Shincheonji * members and those who visited Cheongdo Daenam Hospital * must not stay in the hospital as a guardian.
5. Patients and guardians must limit their movement and leave the ward only if necessary.
6. At the time of admission, patients and guardians must inform the medical staff of their religion (to check for Shincheonji * members), their travel history, and whether they have visited any COVID-19 outbreak areas.
7. Patients and guardians must inform medical staff whether they have been notified for self-isolation during the hospitalization period.
8. Visitors are not allowed during the hospitalization period.
9. Generally, the transfer of patients to another ward is prohibited. (However, a patient can be transferred to another ward if a department change is required.)
10. During the hospitalization period, inpatients are not allowed to go outside. If it is necessary to go outside, then the patient must be readmitted after discharge.

* Mass outbreaks occurred in Shincheonji Church and Cheongdo Daenam Hospital.

**Table 2 healthcare-08-00195-t002:** Rules for patient care to prevent nosocomial COVID-19 infection.

Medical staff must wear a mask at all times and have strict hand hygiene. Rules regarding patient care in the outpatient department: (1) Do not permit patients and guardians to enter the hospital if they are not wearing a mask. (2) Pay attention to patients presenting with fever or respiratory symptoms in the isolated outpatient department. (3) Patients must be asked about their travel history as well as whether they are Shincheonji members * or whether they have visited Cheongdo Daenam Hospital *. Inpatient care rules:
A. Rules for inpatient care and nursing:
(1) Check whether patients and guardians have fever and/or respiratory symptoms on admission. (2) Do not share manometers, thermometers, and BST devices with other teams. (3) Use separate manometers, thermometers, and BST devices for patients with fever and those in isolation. (4) Monitor for fever and respiratory symptoms and notify the doctor about any such symptoms, if present. (5) Check the travel history of patients as well as whether they are Shincheonji * members or have visited Cheongdo Daenam Hospital *. If yes, then report this to the manager. (6) If there is a high likelihood that a patient has COVID-19 on a ward, then the manager must report this to the COVID-19 Situation Room authority. (7) When patients or guardians notify staff that they were ordered to isolate during the hospitalization period, this must be reported to the COVID-19 Situation Room. (8) Generally, the transfer of a patient to another ward is prohibited. If necessary, follow the rules regarding the transfer of patients to another ward. (9) During the hospitalization period, inpatients are not allowed to go outside. If it is necessary to go outside, then the patient must be readmitted after discharge.
B. Care rules for patients and guardians:
(1) Patients and guardians must wear a mask. (2) The number of guardians is limited to 1 per patient. (3) Shincheonji members and those who have visited Cheongdo Daenam Hospital * must be removed from the list of approved caregivers. (4) Patients and guardians must limit their movement and not leave the ward, except for undergoing tests. (5) Visitors are not allowed.

* Mass outbreaks occurred in Shincheonji and Cheongdo Daenam Hospital.

**Table 3 healthcare-08-00195-t003:** Criteria for isolation if close contact has been made with a patient with confirmed COVID-19 (made by the Korea Centers for Disease Control and Prevention).

Criteria	Exposure Risk	Response
Performing a procedure on a patient with confirmed COVID-19 or staying in the same room during the procedure without wearing personal protective equipment (all 4 kinds *). Procedures: cardiopulmonary resuscitation, intubation, bronchoscopy, nebulizer therapy, and aspiration.	High -Medium	Not allowed to work for 2 weeks after the last exposure
Close contact with a patient without wearing personal protective equipment (N95 mask and goggles). This is regardless of whether the patient wore a mask.	Medium	
Medical staff who did not wear gloves and came into direct contact with infected excreta/feces without hand hygiene.	Medium	
Medical staff who wore a mask →Close contact with patients who wore a mask.	Low	Allowed to work but must self-monitor
Medical staff wore personal protective equipment (all 4 kinds *) →Handled or touched excreta/feces.	Low	
Medical staff did not wear personal protective equipment (all 4 kinds *) → movement between wards without contact.	Low	

* 4 kinds of protective equipment: N95 mask, gown, gloves, and goggles or a face shield.

**Table 4 healthcare-08-00195-t004:** Management rule for confirmed cases among the inpatients of Yeungnam University Medical Center.

1. Transfer of patients with confirmed COVID-19 to another ward: (1) Preparations: a negative air-pressure cart, vinyl. (2) Transfer staff: -The medical staff of the corresponding wards (2 people). -Security team (1 person): securing the elevator control and a patient transport path. (3) Gear of transfer staff:		
Medical staff in the ward	Security team	Infection control team
		(when required)
N95 mask, AP gown, gloves, and goggles	N95 mask, AP gown, and gloves	N95 mask, AP gown, and gloves
(4) * If the patient has severe symptoms such as coughing, it is recommended that medical staff in the ward should wear level D gowns.		
2. Transport method:		
(1) Once a room is assigned for the patient to be transferred to, the transfer needs to be arranged with the destination ward when the least patient traffic is expected. (2) Notify the security team (6651) of the respiratory ward on the 1st floor of the transport. (3) The security team goes to the ward with a negative air-pressure cart and a remote control for the elevator. (4) Medical staff in the ward should wear protective equipment (at least an N95 mask, AP gown, gloves, and goggles) and transfer the patient to the negative air-pressure cart. (5) Clean the surface of the negative air-pressure cart with a disinfectant. (6) Once out of the ward, the AP gown and gloves need to be replaced with new ones. (7) The security team secures the transport path for the patient, and the medical staff of the ward moves the cart. (8) After arriving at the destination ward, the patient is handed over in the clean zone.		
3. Transport path:		
(1) Departments	Moving path	
(2) Ward in the Main (3) Building	Elevator No. 2 of the West Building (1st floor in the basement) → Elevator No. 1 of the Respiratory Ward → the corresponding ward	
(4) Intensive Care Unit in the Main Building	Elevator No. 2 for the outpatient department in the Main Building (1st floor in the basement) → Elevator No. 1 of the Respiratory Ward → the corresponding ward	
(5) Wards 130 and 132	Patient elevator in the Main Building (1st floor in the basement) → Elevator No. 1 of the Respiratory Ward → the corresponding ward	
(6) Regional Emergency Medical center & critical care unit	Elevator No. 2 of the West Building (1st floor in the basement) → Elevator No. 1 of the Respiratory Ward → the corresponding ward	
* While the elevator is in use, others are not allowed in.		
4. Measures after transport:		
(1) Take off the protective equipment used during patient transport in the designated dressing room and return to the ward.		
5. Quarantine/disinfection:		
(1) After all patients with confirmed COVID-19 have been moved, the general affairs team needs to be contacted for quarantine. (2) After quarantine, the sanitation team cleans the ward. (3) Clean all medical devices used for testing the patient using a disinfectant (medical disinfectant and disinfecting tissues).		
6. Management of those who came into contact with the confirmed case:		
(1) Investigation period for those who came into contact with the confirmed case: those who came into contact with the confirmed case from the day before the onset of symptoms in the confirmed case: -The investigation can be expanded depending on the movement range of the confirmed case, for example, patients who shared the same space with the confirmed patient.		
(After the investigation of the individuals who were in close contact, the infection control team is informed.)		
7. Management of the close contacts:		
(1) Movement between wards: Generally, patients should be isolated in a single occupancy room. If no single occupancy room is available, cohort isolation in the isolated ward will be used. (2) Filling in the application form for isolation by a doctor.		
8. Management of the close contacts (staff):		
(1) Make a list of the close contacts and consider whether they were wearing protective equipment and the contact time, after the infection control team and the nursing headquarters have checked the CCTV recordings. (2) The infection control team decides whether staff needs to self-isolate or self-monitor.		
9. Rules for preventing contacts.		
(1) Comply with the standards, including wearing a mask and hand hygiene. (2) Those subject to isolation: Xomply with the isolation rule and submit the required paperwork after being informed by the infection control team (isolation notice), monitor symptoms, and undergo a test for COVID-19. (3) Those subject to self-monitoring: Continuously monitor symptoms. Report to the manager or infection control team during symptoms onset. This should be followed by a test or time off work according to the instructions given.		

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
