# Peer review of "Strategies for the Prevention of the Intra-Hospital Transmission of COVID-19: A Retrospective Cohort Study"

_healthcare, 2020, doi:10.3390/healthcare8030195_

Round 1

Reviewer 1 Report

Dear authors,

thank you very much for giving me the opportunity to read your manuscript.

The article describes the rules for avoiding nosocomial infections in a specific hospital and describes the outcome of contact between patients with Covid-19 and other patients or staff in that hospital.

Especially the case study (cases 2 and 5) can be very helpful for other hospitals to improve their hygiene rules.

To improve the article, I have the following suggestions:

  • The methods used should be described in more detail. Which data source was used for which variable. For example, it is conceivable to measure the distance between people using video surveillance (CCTV) or to ask them using a survey. What was used for what is not clear from the article.
  • You should also mention which ethics committee authorized the study.
  • For other information, however, it is unclear why it is needed with such precision e.g. the long list of occupations (line 94-99). I also wonder why you present table 4: The list of measures taken to prevent spreading Covid-19 is impressive, but you do not refer to it in the text/do not make clear what we can learn from the management rules who were not (?) violated.
  • I added minor comments in the PDF attached.

Author Response

The reviewer 1

Dear authors,

thank you very much for giving me the opportunity to read your manuscript.

The article describes the rules for avoiding nosocomial infections in a specific hospital and describes the outcome of contact between patients with Covid-19 and other patients or staff in that hospital.

Especially the case study (cases 2 and 5) can be very helpful for other hospitals to improve their hygiene rules.

To improve the article, I have the following suggestions:

  • The methods used should be described in more detail. Which data source was used for which variable. For example, it is conceivable to measure the distance between people using video surveillance (CCTV) or to ask them using a survey. What was used for what is not clear from the article.

Answer: I appreciate your kind comment for our manuscript. Initially we conducted a questionnaire survey with patients and staff who came into contact with patients with COVID-19. Next, we double checked the results of the survey using patients’ electronic medical records and closed-circuit television (CCTV) recordings. We added this content to the manuscript.

  • You should also mention which ethics committee authorized the study.

Answer: Following the reviewer’s comment, we added the name of ethics committee.

  • For other information, however, it is unclear why it is needed with such precision e.g. the long list of occupations (line 94-99). I also wonder why you present table 4: The list of measures taken to prevent spreading Covid-19 is impressive, but you do not refer to it in the text/do not make clear what we can learn from the management rules who were not (?) violated.

Answer: The contents related with occupations can be helpful for presenting the real situation of the hospital. Also, it would be helpful to know individuals who are at risk for infection of COVID-19. Also, we inserted Table 4 (referred to “transmission path of COVID-19” section) for showing the method how confirmed COVID-19 patients can be transferred to isolation ward for COVID-19 patients. We thought that it would be helpful for other hospital to elucidate the strategy related with this matter. However, to present detail paths for transferring confirmed COVID-19 is not necessary, accordingly we deleted it.  

  • I added minor comments in the PDF attached.

Answer: We appreciate your comments for enhancing the quality of our study. We change “incidence” into “prevalence”. Also, we provided the source of the prevalence rate to the manuscript.

Regarding the memo related with “without further transmission”, the content you wrote is right.

Reviewer 2 Report

The paper submitted by the authors appears clear and very useful for comparing the risk containment procedures of covid-19 infection implemented in hospitals around the world.
Only some minor remarks regarding in particular the tables and figures.
- please specify if hospital differentiated access for covid or suspected covid patients and for other patients in the periodo considered. specify if the data refer to non-covid wards or to covid wards.
- Figure 1. I wonder why there are some confirmed patients who have not come into contact with any health worker. Please specify. 
- Tables 1 and 2 seem to overlap partially. Does Table 1 summarize the common rules for inpatient and outpatient patients, guardians and medical staff? if so, you could consider to retain only table 2.
- table 4 point 5. It does not seem to me necessary to indicate specific paths but only that there were specific and differentiated paths.

Author Response

The reviewer 2

The paper submitted by the authors appears clear and very useful for comparing the risk containment procedures of covid-19 infection implemented in hospitals around the world.
Only some minor remarks regarding in particular the tables and figures.
- please specify if hospital differentiated access for covid or suspected covid patients and for other patients in the periodo considered. specify if the data refer to non-covid wards or to covid wards.

Answer: I appreciate your kind comments for the manuscript. The hospital used different wards for patients who contacted with a confirmed patient, patients confirmed COVID-19 patients, and general patients not related to COVID-19. We added this content to the manuscript.

- Figure 1. I wonder why there are some confirmed patients who have not come into contact with any health worker. Please specify. 

Answer: We specified the reason for above comment as follows.

*Case 15, 18, 20, 22, and 29 were isolated immediately after they arrived at the hospital because they have symptoms related with COVID-19.

- Tables 1 and 2 seem to overlap partially. Does Table 1 summarize the common rules for inpatient and outpatient patients, guardians and medical staff? if so, you could consider to retain only table 2.

Answer: Table 1 is related with rules for inpatient and outpatient patients and patients’ guardians, and Table 2 is for rules for medical staff. Therefore, two tables contain different contents and lessons. Therefore, we think it would be better not to delete one of them.

- table 4 point 5. It does not seem to me necessary to indicate specific paths but only that there were specific and differentiated paths.

Answer: I agree with your comment. Specific path is not necessary. We deleted the contents related with the comment.